# Oral glucose tolerance test clearance in type 2 diabetes patients who underwent remission following intense lifestyle modification: A quasi-experimental study

**Pramod Tripathi[1], Nidhi Kadam[1]\*, Diptika Tiwari[1], Anagha Vyawahare[1], Baby Sharma[1], Thejas Kathrikolly[1], Maheshkumar Kuppusamy[2], Venugopal Vijayakumar[3]**

1 Department of Research, Freedom from Diabetes Research Foundation, Pune, Maharashtra, India,
2 Department of Physiology, Government Yoga and Naturopathy Medical College and Hospital, Arumbakkam, Chennai, Tamil Nadu, India, 3 Department of Yoga, Government Yoga and Naturopathy Medical College and Hospital, Chennai, Tamil Nadu, India

\* research@freedomfromdiabetes.org

**Data Availability Statement:** All the data are in Supporting Information files named as Data Set.

## Abstract

Achieving diabetes remission (HbA1c<48mmol/mol without the use of anti-diabetic medication for 3 months) might not assure restoration of a normal glycemic profile [fasting blood sugar level <5.6 mmol/L and Post-Prandial (PP) blood glucose <7.8mmol/L]. The study investigates the factors associated with OGTT clearance in patients under type 2 diabetes remission. Four hundred participants who achieved remission during a one-year online structured lifestyle modification program, which included a plant-based diet, physical activity, psychological support, and medical management (between January 2021 and June 2022), and appeared for the OGTT were included in the study. OGTT clearance was defined by fasting blood glucose < 5.6 mmol/L and 2-hour post-prandial blood glucose <7.8 mmol/L post-consumption of 75g glucose solution. Of the 400 participants, 207 (52%) cleared OGTT and 175 (44%) had impaired glucose tolerance (*IGT*). A shorter diabetes duration (<5 years) was significantly associated with OGTT clearance (p<0.05). Pre-intervention use of glucose-lowering drugs showed no association with OGTT clearance (p<0.1). Post-intervention, the *OGTT-cleared* group showed significantly higher weight loss (p<0.05) and a decrease in HbA1c compared to the *IGT* group (p<0.05). Improvement in Insulin resistance and β-cell function was also higher in the *OGTT-cleared* group compared to the *IGT* group (p<0.05). In conclusion, clearing the OGTT is a possibility for those achieving remission through lifestyle interventions. Higher weight loss, a shorter duration of diabetes, and improvement in insulin resistance were significantly associated with OGTT clearance in participants in remission. Future randomized controlled trials with longer follow-ups may help substantiate our findings.

## Introduction

Type 2 diabetes (T2D) is a complex condition primarily characterized by insulin resistance, impaired pancreatic islet cell function, elevated metabolic irregularities, and disturbances in

**Funding:** The author(s) received no specific funding for this work.

**Competing interests:** The authors declare no competing interests.

glycemic control [1]. It can lead to several complications, some of which may result in fatality due to uncontrolled glycemic levels [2]. Previous research has shown that improved glycemic control and remission can be achieved through lifestyle changes such as weight loss, low-calorie diet, and physical activity [3–6]. The American Diabetes Association (ADA) has recommended an assessment of psychosocial and emotional health concerns in patients with diabetes, making it an integral part of holistic diabetes care in addition to conventional lifestyle changes [7–9].

Diabetes remission is defined as HbA1c levels below 48 mmol/mol without the use of pharmacotherapy for more than 3 months [10]. Long-term remission in T2D-diagnosed patients is most predicted by sustained weight loss, shorter duration of diabetes, and reduced insulin resistance, in addition to improved pancreatic function [11, 12] Multiple studies have suggested that long-term maintenance of remission and prevention of relapse in T2D is mostly achieved by long-term adherence to lifestyle therapy and weight loss, provided β-cell dysfunction is not irreparable [4, 5, 13].

The oral glucose tolerance test (OGTT) is widely regarded as the gold standard for diagnosing T2D because of its ability to effectively assess a patient's glucose processing capabilities [14]. It is also commonly used to screen for diabetes mellitus, insulin resistance, impaired pancreatic β-cell function, reactive hypoglycaemia or acromegaly, and rare carbohydrate metabolic diseases [15]. While most studies on diabetes remission are based on HbA1c levels, the OGTT remains one of the most essential diagnostic tools for assessing normal glucose profiles [16, 17]. However, there is a lack of studies exploring the possibility of achieving a normal glycaemic profile in terms of OGTT clearance post-T2D remission. The current study aimed to investigate the factors associated with OGTT clearance in T2D patients who achieved remission after attending an online one-year lifestyle intervention program. The study hypothesised that T2D-diagnosed patients in remission after customized lifestyle intervention may be able to attain a normal glycemic profile through OGTT clearance, in addition to lowering HbA1c levels and improving insulin-related factors [Homeostatic model assessments of insulin resistance (HOMA-IR) and β-cell function (HOMA-β)].

## Materials and methods

### Study design & participant enrolment

The present study is a quasi-experimental with a pre-post design. Participants were recruited from the Freedom from Diabetes Clinic in Pune, India. Of the 7839 T2D patients who enrolled for the one-year online lifestyle modification program (between January 2021 and June 2022), 3689 achieved remission and were invited to participate in a 3-month 'glycemic ladder' training as part of the one-year program. Of these, 1121 participants accepted the invitation and underwent a 3-month glycemic training, which included gradual introduction of foods with an increasing glycemic index every week while monitoring blood glucose levels. Four hundred participants who completed the three-month training, fulfilled the remission criteria, and underwent the oral glucose tolerance test (OGTT) were included in the final analysis. This study was approved by the Institutional Ethics Committee (Ref. No. ECR/45/Indt/MH2013/RR-16). Written informed consent was obtained from all participants.

### Inclusion criteria

The primary inclusion criterion was 'remission' defined as 'HbA1c less than 48 mmol/mol without any glucose-lowering medicines (including alternative medicine) for at least 3 months', achieved during a one-year lifestyle intervention in addition to completion of 3-month glycemic ladder training and attempting OGTT.

## Intervention

Participants enrolled in the one-year lifestyle intervention program were assigned a team of six experts, including a physician, nutritionist, physiotherapist, psychologist, mentor (a previous program participant who benefited from the program and volunteers to guide new participants), and monitor (for follow-ups and reminders for lab tests and appointments). Participants were instructed to follow the Freedom from Diabetes protocol, a one-year program divided into four phases focusing on four core protocols: diet, exercise, psychological support, and medical management. Each participant had access to a dedicated mobile application that allowed them to communicate with a team of experts via voice, video, and text. The entire program was delivered through educational group sessions (12 sessions- one every month) and individualized guidance through a team of experts.

The ethical framework of our intervention emphasized the significance of individualized dietary modifications, customized exercise schedules, yoga, and stress management techniques, while simultaneously acknowledging the cultural nuances associated with dietary preferences and exercise choices. Our approach offered participants the flexibility to select from a range of options while still adhering to the primary objective of the intervention.

The dietary intervention initially focused on a plant-based diet. The dietary modification was individualized based on the BMI and associated medical conditions of the participants. A strict vegan diet was recommended for the first 2 months. In the first month, overall calorie intake was reduced to an average of 1100 kcal/day. In the second month, calorie intake was further reduced (400–500 kcal/day on the day of fasting) through intermittent fasting and juice fasting to aid in faster weight loss, depending on the BMI of the participant; those with normal BMI did not undergo juice and water fasting. Once the target BMI (23–25 kg/m$^2$) was achieved, calorie intake was gradually increased to 1800–2000 kcal to promote muscle building (with the aid of exercises). Once the blood sugar levels stabilized, other food items from the regular diet were reintroduced phase-wise under the close supervision of the nutritionist and physician.

The exercise intervention included exercises to promote blood circulation and muscle activation (warm-up, Surya Namaskar, super brain yoga, palm planks), with a major focus on improving strength, flexibility, and stamina by adopting the 3-2-1 pattern per week (3 hrs. of strength, 2 hrs of flexibility and 1 hrs. of stamina exercises). A customized exercise plan (e.g., swimming, running, cycling, yoga) was suggested to the participants based on their age and preference to create a consistent level of fitness by the end of the program.

The third protocol of the psychological intervention focused on understanding one's stress and anxiety levels and raising awareness in participants about the mind-body connection through online group sessions. Participants were advised to manage their stress through journaling and meditation. Individual counselling was also provided to the participants by trained psychologists who used specific methods such as cognitive behaviour therapy (CBT), rational emotional behaviour therapy (REBT), neuro-linguistic programming (NLP), clinical hypnotherapy, life coaching, and *Pranic healing*, as required.

The final protocol of the intervention was the medical protocol delivered by the assigned physician and consisted of daily drug-dose adjustments based on blood sugar level updates on the mobile application in addition to correcting nutritional deficiencies identified through biochemical tests (performed every 3 months). Once blood sugar levels stabilized, participants were asked to report their blood sugar levels less frequently (weekly/fortnightly).

As a part of the one-year lifestyle intervention program, upon achieving remission, participants were asked to voluntarily enrol for a three-month 'glycemic-ladder' training where they were gradually introduced to food items that were restricted during the intervention. Items

such as grains for breakfast, fruits, and desserts were reintroduced into the diet while closely monitoring their blood sugar levels followed by increasing doses of glucose (15g, 25g, 30g, 40g, 50g, 60g, and 75g in 250 ml water) to gradually increase glucose tolerance. All the above 'meal/ glucose experiments' were performed once a week, preferably on weekends, at a predetermined time. Throughout the training, participants were requested to monitor and report their blood glucose levels in close consultation with their nutritionist and physician. The participants were also advised to avoid intense exercise on the day of the experiments. After the last experiment with 75g of glucose, they were asked to take a break of 15 days while continuing their routine recommended diet and exercise regimen before attempting the final OGTT. Participants who completed and cleared the training with an HbA1c level of less than 48 mmol/ mol without glucose-lowering medication were considered eligible for the final oral glucose tolerance test.

Adherence to the overall program was monitored through 12 structured questionnaires administered monthly through our mobile application. This ensured the engagement of participants and provided direct feedback to all respective experts to take necessary action to ensure adherence to the protocol.

## Assessment of anthropometric and biochemical parameters

Data on anthropometric measurements (height and weight), biochemical parameters (HbA1c, fasting blood sugar, post-prandial blood sugar, fasting insulin), and medical history (diabetes duration and medication status) were assessed at baseline. While weight and fasting and, post-prandial blood glucose levels were assessed daily until the target BMI was achieved (23–25 kg/ m$^2$) or blood glucose levels stabilized, HbA1c was repeated every 3–4 months and at the time of OGTT. Homeostatic model assessment of insulin resistance (HOMA-IR) and β-cell function (HOMA-β) at baseline and at the end of one year were calculated using standard formulas [18].

## Oral glucose tolerance test (OGTT)

During the OGTT, a fasting blood sample was obtained to establish the baseline blood glucose levels. The participants then consumed 75g of dextrose monohydrate in 250 ml of water (WHO-recommended dosage for OGTT) [19]. After 120 minutes (2 hrs) of glucose consumption, a second blood sample was collected to determine the post-prandial (PP) blood glucose level. Following the OGTT, the participants were classified as *OGTT-cleared* [fasting blood sugar level (FBSL) <5.6 mmol/L and PP blood glucose level <7.8mmol/L], *IGT* (Impaired Glucose Tolerance) [FBSL 5.6–6.9 mmol/L and PP blood glucose level 7.8-11mmol/L], and *Not-cleared* (FBSL ≥7.0 mmol/L and PP blood glucose level ≥11.1mmol/L), based on ADA guidelines [20].

## Statistical analysis

Statistical analyses were performed using IBM SPSS version 21.0. Data are presented as frequencies and percentages for categorical variables. Mean ± standard deviation and median (interquartile range) were reported for continuous variables with normal and skewed distributions, respectively. The absolute percentage change was calculated as the final value—initial value/initial value x100. Parametric and non-parametric (ANOVA and Kruskal Wallis) tests were used to test the significance of the differences between groups based on the distribution of data. To compare pre-and post-intervention measurements, the Wilcoxon signed-rank test was used for skewed data whereas paired t-tests and independent t-tests were used to analyze normally distributed data. Statistical significance was set at $p < 0.05$.

## Results

The results presented here reflect the outcomes observed after a one-year lifestyle intervention program for diabetes management. The participants were subjected to a comprehensive one-year intervention that included dietary, exercise, psychological and medical management. The study findings highlight the impact of the customized lifestyle intervention on OGTT clearance in participants under remission.

### General characteristics of the participants

The mean age (years) of the participants during OGTT was 51.7±1.7 and most participants were male (59%). At the time of joining the one-year intervention, the median duration (interquartile range) of diabetes was 4.3 (1.3–8.8) years and most participants (84.3%) were on glucose-lowering medication [including oral hypoglycemic agents (OHAs) or insulin or both]. Overall, the mean duration of remission at the time of attempting OGTT was 4.6 ± 1.9 months.

### Oral glucose tolerance test (OGTT)

All four hundred participants under remission attempted the OGTT as part of the program. Of these, 207 (52%) cleared the oral glucose tolerance test (OGTT), 175 (44%) showed impaired glucose tolerance (IGT), and 18 (4%) were not able to clear the OGTT. At the time of appearing for OGTT, the mean weight (kg), median BMI (kg/m$^2$), and HbA1c (mmol/mol,) of the participants were 65.3±9.8, 23.5 (21.9–25.3) and 39.8 (36.6–43.1), respectively. The mean time taken to achieve remission before starting glycemic ladder training for the 3 groups was: OGTT cleared group, 4.4 ± 2.0 months; IGT group, 4.7 ± 1.8 months; Not cleared group, 5.6 ± 2.5 months.

The sociodemographic profiles of the participants in the three groups are presented in Table 1. All three groups had similar sociodemographic characteristics (P >0.1). Diabetes

**Table 1. Socio-demographical profile of participants in three outcome groups (N = 400).**

| Parameters | | OGTT Status | | |
|---|---|---|---|---|
| | | *OGTT-cleared* (N = 207) | *IGT* (N = 175) | *Not-cleared* (N = 18) |
| Age (Years) | | 52.0±9.4 | 51.1±10.0 | 52.8±9.8 |
| Gender | Male | 116 [56.0] | 107 [61.1] | 12 [66.6] |
| | Female | 91 [44.0] | 68 [38.9] | 6 [33.4] |
| Education | Graduate and below | 109 [52.7] | 98[56.0] | 12 [66.7] |
| | Post-Graduate and above | 98 [47.3] | 77[44.0] | 6 [33.3] |
| Occupation | Employed | 118 [57.0] | 113 [64.6] | 13 [70.6] |
| | Retired | 28 [13.5] | 19 [10.8] | 3 [17.6] |
| | Homemaker | 39 [18.8] | 32 [18.3] | 2 [11.8] |
| | Others [a] | 22 [10.6] | 11 [6.3] | - |
| Marital Status | Married | 184 [88.8] | 156 [89.1] | 17 [94.1] |
| | Single [b] | 23 [11.2] | 19 [10.9] | 1 [5.9] |
| Family History | Yes | 160 [77.1] | 133 [76.0] | 14 [76.5] |
| | No | 47 [22.9] | 42 [24.0] | 4 [23.5] |

Data are presented as mean ± standard deviation or frequency [percentage]

[a] Student/Preferred not to disclose

[b] Never married/separated/widowed; No significant differences were observed between the groups by Chi-square test (p>0.1)

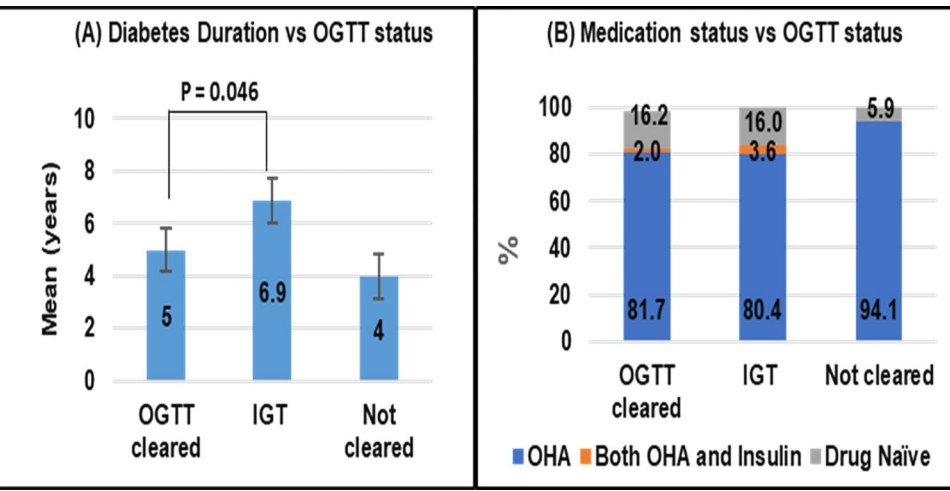

**Fig 1.** Group-wise comparison (*OGTT-cleared* n = 207, *IGT* n = 175 and *Not-cleared* n = 18) of diabetes duration (A) and medication status (B).

duration was significantly lower in the OGTT-cleared group than in the IGT group (p = 0.046) (Fig 1A). Due to the low number of patients who did not clear the OGTT (N = 18), we could not statistically compare the outcome for this group. No significant difference in OGTT results was observed based on initial medication status (p>0.1) (Fig 1B).

Changes in anthropometric and biochemical parameters post-one-year lifestyle intervention in three outcome groups (*OGTT-cleared*, *IGT and Not-cleared*) have been described in Table 2.

Post-one-year intervention assessment showed significant improvement in weight, BMI, HbA1c, fasting insulin, fasting blood sugar, post-prandial blood sugar, HOMA-IR, and HOMA-β in all the participants who cleared OGTT and showed impaired glucose tolerance

**Table 2. Changes in anthropometric and biochemical parameters post-one-year lifestyle intervention (N = 400).**

| Parameters | Groups Based on OGTT Status | | | | | | | | |
|---|---|---|---|---|---|---|---|---|---|
| | *OGTT-cleared* (N = 207) | | P values | *IGT* (N = 175) | | P values | *Not-cleared* (N = 18) | | P values |
| | Pre-Intervention | Post-Intervention | | Pre-Intervention | Post-Intervention | | Pre-Intervention | Post-Intervention | |
| **Weight (kg)** | 76.4±13.6 | 66.0±10.4 | <0.001 | 71.4±11.1 | 64.6± 9.2 | <0.001 | 72.2±10.9 | 66.9±11.3 | <0.001 |
| **BMI (kg/m²)** | 27.8 (24.6–31.3) | 23.8 (21.8–25.8) | <0.001 | 26.0 (23.0–28.3) | 23.1 (21.9–25.3) | <0.001 | 25.8 (24.8–26.9) | 23.4 (22.7–25.2) | 0.002 |
| **HbA1c (mmol/mol)** | 55.2±17.2 | 38.1±4.6 | <0.001 | 56.1±18.0 | 41.1±5.1 | <0.001 | 52.2±9.2 | 44.7±4.4 | 0.001 |
| **Fasting Insulin (pmol/L)** | 52.1 (33.6–81.4) | 36.6 (25.7–53.4) | <0.001 | 52.1 (36.3–76.7) | 40.7 (28.2–60.0) | <0.001 | 61.6 (43.9–92.1) | 65.4 (40.4–75.0) | 0.127 |
| **Fasting Blood Sugar (mmol/L)** | 6.5±1.9 | 5.2±0.73 | <0.001 | 6.9±1.9 | 5.9±0.79 | <0.001 | 6.5±1.0 | 6.1±0.67 | 0.122 |
| **PP-Blood Sugar (mmol/L)** | 7.1±2.6 | 6.2±1.9 | <0.001 | 7.7±2.7 | 6.9±1.9 | 0.001 | 7.8±2.2 | 7.8±2.1 | 0.979 |
| **HOMA-IR** | 2.4 (1.4–4.0) | 1.4 (1.0–2.1) | <0.001 | 2.7 (1.7–3.8) | 1.7 (1.1–2.7) | <0.001 | 3.3 (2.5–5.5) | 2.8 (1.8–3.4) | 0.044 |
| **HOMA-β** | 62.7 (41.3–103) | 72.6 (48.7–113) | <0.001 | 55.7 (35.0–81.0) | 58.3 (39.4–86.9) | 0.357 | 79.4 (54.8–141) | 92.1 (46.4–114) | 0.877 |

Data are presented as mean ± standard deviation or median (Interquartile range)

OGTT: Oral Glucose Tolerance Test; IGT: Impaired Glucose Tolerance; BMI: Body mass index; PP: Post-Prandial; HbA1c: glycosylated haemoglobin; HOMA-IR: Homeostatic model assessment of Insulin resistance (HOMA-IR); HOMA-β: Homeostatic model assessment of β-cell function

**Table 3. Between-group comparison of percentage change in anthropometric and biochemical parameters post-one-year intervention.**

| Parameters | Percentage Change | | |
|---|---|---|---|
| | *OGTT-cleared* | *IGT* | *Not-cleared* |
| Weight (Kg) | (-12.9±7.7) [a, b] | (-9.0± 6.8) | (-7.4±5.1) |
| BMI (Kg/m²) | [-13.3 (-19.0 – -8.0)] [a, b] | [-9.2 (-14.9 – -3.7)] | [-5.9 (-9.6 – -2.6)] |
| HbA1c (mmol/mol) | (-26.5±18.7) [a, b] | (-21.9 ± 19.8) [b] | (-11.7±17.9) |
| Fasting Insulin (pmol/L) | [-32.1 (-55.6–0.10)] | [-18.5 (-45.4–14.8)] | [-7.6 (-36.0–12.8)] |
| Fasting Blood Sugar (mmol/L) | (-14.1±19.8) | (-11.6±17.4) | (-4.7±13.9) |
| PP-Blood Sugar (mmol/L) | (-7.5±32.7) | (-6.2±32.4) | (7.2±40.1) |
| HOMA-IR | [-37.1 (-65.8 – -8.0)] [a] | [-22.8 (-52.7–2.9)] | [-13.8 (-49.2–-4.1)] |
| HOMA-β | [11.8 (-23.5–79.3)] | [0.53 (-26.4–50.0)] | [-4.5 (-30.1–29.6)] |

Data for percentage change is presented as Mean ± Standard Deviation or median (Interquartile range)

OGTT: Oral Glucose Tolerance Test; IGT: Impaired Glucose Tolerance; BMI: Body mass index; PP: Post-Prandial; HbA1c: glycosylated haemoglobin; HOMA-IR: Homeostatic model assessment of Insulin resistance (HOMA-IR); HOMA-β: Homeostatic model assessment of β-cell function

The percentage decrease in weight, BMI, HbA1c, Fasting Insulin, Fasting Blood Sugar, PP-Blood Sugar, and HOMA-IR along with a percentage increase in HOMA-β indicates a positive effect and benefit to the participants

[a] Significantly different than *IGT*

[b] significantly different than the *Not-cleared* group at p-value <0.05

(IGT) except for HOMA-β in *IGT* group. Additionally, the *Not-cleared* group showed significant improvement in weight, BMI HbA1c and HOMA-IR; notably the post-intervention values for HOMA-IR remained outside the normal range (<2.5) [21] (**Table 2**).

## Comparison of anthropometric and biochemical profiles between groups after one year of lifestyle intervention

The group-wise analysis based on OGTT status showed that percentage change in weight (p = <0.001), BMI (p<0.001), HbA1c (p = 0.025), and HOMA-IR (p = 0.023) was significantly higher in the *OGTT-cleared* group when compared to the *IGT* group. Moreover, the *OGTT-cleared* group showed significantly higher percentage changes in weight (p = 0.003), BMI (p = 0.004), and HbA1c (p = 0.003) compared to the *Not-cleared* group. Furthermore, the *IGT* group showed a significantly higher percentage change in HbA1c (p = 0.039) compared to the *Not-cleared* group. It's crucial to note that the sample size in the *Not-cleared* group was insufficient for a robust statistical analysis (**Table 3**).

## Discussion

The present study highlights the likelihood of achieving a normal glycemic profile (FBSL <5.6 mmol/L and PP blood glucose < 7.8 mmol/L) in terms of clearing OGTT in participants undergoing T2D remission following a one-year holistic lifestyle intervention program. Compared to OGTT which is more reliable as a screening tool for diabetes, an HbA1c of less than 48 mmol/mol neither predicts nor screens the normal glucose level in those at high risk for T2D [16, 22]. We tested the hypothesis that those achieving remission can achieve a normal glycemic profile post-lifestyle modification and observed that around 50% did achieve it with another 44% reaching the pre-diabetes stage.

The participants were able to achieve and maintain significant weight loss and a drop in HbA1c at the end of the one-year intervention, emphasising the significance of weight loss and a drop in HbA1c for remission. These results are consistent with earlier reports on lifestyle

interventions, showing that sustained weight loss and lower HbA1c levels are crucial for remission [23].

Further analysis to determine factors associated with a return to normal glycemic profile indicated that a shorter diabetes duration was significantly associated with OGTT clearance. This is consistent with earlier findings in which shorter diabetes duration was cited as one of the critical predictors of T2D remission [24, 25]. Since pharmacotherapy cessation is one of the mandates for remission, we examined the role of baseline (pre-intervention) medication status in the clearance of OGTT. Pre-intervention pharmacotherapy showed no significant association with OGTT clearance. This highlights that even those on multi-drug therapy have the potential to attain remission through lifestyle intervention.

The pathogenesis of diabetes and pre-diabetes is substantially impacted by insulin resistance and pancreatic β-cell dysfunction [26, 27]. In the present study, we assessed how our holistic lifestyle intervention affected insulin resistance (HOMA-IR) and β-cell function (HOMA-β) in T2D-diagnosed participants who had achieved remission. Our results indicate significant improvement in insulin resistance (HOMA-IR < 2.5) and β-cell function post-intervention compared to baseline in all the participants who appeared for OGTT. Our findings are consistent with the HOMA-IR results but not with the HOMA-β, as reported by **Gravel et al. (2015)** [28] where their lifestyle intervention which included High-intensity Interval Training (HIIT) and Mediterranean diet, showed significant improvement in insulin resistance but no significant improvement in β-cell function after a 9-month intervention. Additionally, the improvement in both HOMA-IR and HOMA-β was significantly higher in the *OGTT-cleared* group compared to the *IGT* group. This observation is in line with the study by **Kahn et al 2014** [29] who reported that insulin resistance and loss of β-cell activity are critical in the development and progression of impaired glucose control due to elevated glucose levels. Despite low numbers, a substantial reduction in insulin resistance (HOMA-IR) was observed post-intervention in the *Not-cleared* group, yet the results remained suboptimal (HOMA-IR = 2.8); concurrently there was a decrease in HOMA-β, contradicting **Kahleova et al.'s** 2019 [30] reports of improved beta-cell function and fasting insulin sensitivity in a 16-week dietary intervention for overweight individuals without diabetes history. The intricate interplay of genetic and age-related factors, baseline metabolic status, and diabetes duration may contribute to the observed variations [13, 31]. The small sample size poses a challenge in drawing definitive conclusions, due to the inherent variability in our study population for the *Not-cleared* group, emphasizing the need for cautious interpretation. Future research with a larger sample size could delve into the specific mechanisms underlying the observed changes, concerning HOMA-IR and HOMA-β. However, our study implies that with lifestyle intervention, it may be possible to improve insulin resistance along with β- cell function and achieve remission with a normal glucose profile, provided that there is no permanent damage to β-cells as suggested by **Nagi et al. 2019** [5].

Adhering to the lifestyle intervention protocol and sustaining weight loss is necessary for maintaining the state of remission with a normal glycemic profile [4, 32]. Significant weight loss can improve hepatic insulin sensitivity and redifferentiation of β-cells, resulting in a β-cell recovery in the pancreas in T2D patients [33]. These findings are consistent with our study, where we observed that post-intervention all three groups showed sustained weight loss. However, participants who cleared OGTT had significantly higher weight loss compared to the *IGT* group, and they were able to sustain the weight loss at the end of one year with improved HOMA-IR and HOMA-β. We also observed a significant and sustained drop in the HbA1c levels for the *OGTT-cleared group* compared to the *IGT* group. Over half of the participants with shorter diabetes duration achieved remission by attaining a normal glycemic profile, including a fasting plasma glucose level of 5.6 mmol/L, a post-prandial plasma glucose level of

7.8 mmol/L, and an HbA1c level of less than 39 mmol/mol. This was achieved through sustained weight loss and improved HOMA-IR and HOMA-β function post-one-year lifestyle intervention. We previously published similar findings in a case series in which four T2D-diagnosed participants under remission were able to clear OGTT consecutively for three years, implying the possibility of long-term remission through lifestyle changes [34]. Irrespective of OGTT status, our data showed that at the end of the one-year lifestyle intervention program, all the participants were able to maintain the weight loss and drop in HbA1c level and continued to be in remission, indicating the sustainability of the intervention.

As far as our knowledge extends, no previous research has investigated the remission of Type 2 diabetes with a focus on attaining a normal glucose profile, as assessed through an OGTT following a customized lifestyle intervention. However, our study had certain limitations. In the absence of a control group, the generalizability of our findings to broader populations or contexts was limited. Nevertheless, given the ethical concerns that would arise from withholding the intervention in a real-world setting, this shortcoming may be overlooked. Further, this was a quasi-experimental pre-post study design which reduces the generalizability of the results. On the other hand, it allowed us to study the effect of the intervention as it naturally occurs in real-world settings. Additionally, due to the holistic nature of the intervention with many factors at play, we could not study the isolated effect of any one component. Despite being a limitation, our research highlights the significant effectiveness of the lifestyle management program in achieving a better outcome in T2D management compared to a single intervention. The importance of holistic lifestyle intervention has been widely reported in several other studies, which highlights the importance of our findings in contributing to the body of knowledge in T2D management. Another limitation is the selection bias due to the voluntary nature of participation in our lifestyle management program. Those who choose to participate may be more motivated or have unique traits when compared to the broader diabetes community. This self-selection could impact the external validity of our findings. Although the results of our lifestyle intervention program may not generalize to the broader population with diabetes, this does not diminish the importance of our findings within a specific group of participants.

## Conclusion

More than half of the participants in diabetes remission were able to clear the OGTT post-one-year lifestyle intervention indicating the ability to achieve normal glycemic profile. Participants who cleared OGTT showed shorter diabetes duration, sustained weight loss and a reduction in HbA1c levels. Additionally, improvements in insulin resistance markers suggest their role in increasing the chances of achieving normal glycemic control in patients with T2D. Future randomized controlled trials with longer follow-up periods may substantiate our findings.

## Supporting information

**S1 Dataset.**
(XLSX)

## Acknowledgments

The authors would like to thank the staff of Freedom from Diabetes. Furthermore, we are grateful to all the participants who agreed to participate in the study and gave their consent to use the data for analysis.

## Author Contributions

**Conceptualization:** Pramod Tripathi, Nidhi Kadam, Thejas Kathrikolly.

**Data curation:** Diptika Tiwari, Anagha Vyawahare, Baby Sharma.

**Formal analysis:** Diptika Tiwari, Anagha Vyawahare, Baby Sharma.

**Methodology:** Diptika Tiwari, Baby Sharma, Thejas Kathrikolly.

**Project administration:** Nidhi Kadam.

**Supervision:** Pramod Tripathi.

**Writing – original draft:** Diptika Tiwari, Anagha Vyawahare, Maheshkumar Kuppusamy, Venugopal Vijayakumar.

**Writing – review & editing:** Baby Sharma, Thejas Kathrikolly, Maheshkumar Kuppusamy, Venugopal Vijayakumar.

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
