## [Decision Letter · Decision Letter 0]

13 Nov 2023

PONE-D-23-27928Oral Glucose Tolerance Test Clearance in Type 2 Diabetes Patients who underwent Remission following Intense Lifestyle modification: A Quasi-Experimental StudyPLOS ONE

Dear Dr. Kadam,

Thank you for submitting your manuscript to PLOS ONE. After careful consideration, we feel that it has merit but does not fully meet PLOS ONE’s publication criteria as it currently stands. Therefore, we invite you to submit a revised version of the manuscript that addresses the points raised during the review process.

All the comments should be clearly addressed This manuscript requires a major revisionPlease address all the comments below==============================

We look forward to receiving your revised manuscript.

Kind regards,

Fredirick Lazaro mashili, MD, PhD

Academic Editor

PLOS ONE

Additional Editor Comments:

This study addresses a crucial aspect of diabetes management — the ability to achieve normal glycaemic profiles post-remission through lifestyle interventions. This is a significant area of research with potential implications for diabetes care. Additionally, the intervention includes various elements like diet, exercise, psychological support, and medical management, which collectively contribute to a holistic treatment approach. Furthermore, the authors have collected a wide range of data points, including HbA1c levels, insulin resistance, β-cell function, and anthropometric measures, providing a robust basis for analysis. Despite all these pros, this manuscript could be improved by considering the following.

1. The study does not isolate the effects of individual components of the lifestyle intervention, which could be insightful for understanding which elements are most effective. Discussing this could also improve the manuscript.

2. While the authors have reported statistical significance, the effect sizes of the interventions could be more explicitly stated to understand the magnitude of the changes observed.

3. The introduction could more explicitly state the study’s hypotheses or research questions.

4. The manuscript should address any ethical considerations, particularly given the intervention's comprehensive nature and the potential impact on patients' lifestyles.

5. There may be an element of selection bias, as patients who opt into such a program might be more motivated or have different characteristics than the general diabetic population. This should be discussed.

Reviewers' comments:

Reviewer's Responses to Questions

**Comments to the Author**

1. Is the manuscript technically sound, and do the data support the conclusions?

Reviewer #1: Partly

Reviewer #2: Partly

2. Has the statistical analysis been performed appropriately and rigorously? 

Reviewer #1: No

Reviewer #2: I Don't Know

3. Have the authors made all data underlying the findings in their manuscript fully available?

Reviewer #1: No

Reviewer #2: Yes

4. Is the manuscript presented in an intelligible fashion and written in standard English?

Reviewer #1: Yes

Reviewer #2: Yes

5. Review Comments to the Author

Reviewer #1: I thank the authors for this relevant and informative study. However the authors have overlooked some key issues as noted below.

1. The authors report that their study is a quasi-experimental pre-post study design, however some key data is not clearly depicted.The authors report baseline, post intervnetion and post OGTT data for HOMA-IR and HOMA-B (table 1 and Figure 2), however given they report 3 specific outcome groups (cleared, IGT and not cleared) the baseline and post intervention data should also be grouped as such. There is quite a wide range in the measurements for both HOMA-IR and HOMA-B and it would be easier to discuss the observations noted if the data is grouped as such.

The authors also use post- intervention in table 2 is confusing. Do they mean post the 1 year intervention or post the 3 months of glycaemic-ladder’ training? The table heading should be more clear.

2. The authors also report P values showing significance differences between the groups, however these are only mentioned in the text and in broad terms. The P values are not depicted in any of the figures, and the figures are also lacking in the number of participants for each observation. If the authors were also able to group the baseline and post intervention data into the three observations noted they would be able to compare differences in these parameters that could predict the 3 observations in the OGTT.

3. Although the authors do report that the not cleared group is small (n = 18), the results in this group warrant more exploration and discussion. In this group they observed improvements in HOMA-IR post intervention however there was a decline in beta cell function as noted by HOMA-B measurements. In a previous diet intervention study of much lower duration (A Plant-Based Dietary Intervention Improves Beta-Cell Function and Insulin Resistance in Overweight Adults: A 16-Week Randomized Clinical Trial, doi: 10.3390/nu10020189), improvement in beta cell function were associated with improvements in HOMA-IR as is to be expected. The authors should discuss their unique results in more detail and offer plausible explanations for their observations.

Reviewer #2: General comment: You have a nice study if the following are included in your revision

Title: Okay

Abstract:

-Does not highlight type of lifestyle modification intervention that was offered

Introduction:

-what was the problem here?

-What was you testing? Effectiveness of an intervention or suitability of OGTT in relation to HbA1c? state clearly

Patient Enrolment:

-Why not call them participants?

-Do you think the nature of your sampling plan could have influenced your conclusion?

-do you think the potential participants who achieved remission differed from those who didn’t? and what about those who were not included in the final analysis? Would the results and hence conclusion been different if all who achieved remission had participated? Why?

Inclusion Criteria:

- did you account for potential contamination from alternative glucose lowering therapies such as herbs which could have not been part of the intervention?

Intervention:

-What was the intervention? One year lifestyle modification versus 3-month glycaemic training.

-why were all participants kept under the same calories target at a given stage? Why was the target not individualized depending on personal characteristics? Do you think that could have influenced your results?

-how did you take account non-exercise physical activities such as those related to work?

-how was compliance to intervention ensured and assessed?

-Psychological support was part of intervention but you say nothing in the introduction

-why interventions appears here but nothing in results?

-Participants had individualized exercise sessions, how are they comparable to each other?

Results:

-What was the composition of weight lost (water, fat or lean mass)? How was is measured? Do you think it could have influenced the results?

-Pre-intervention BMI range (21.4-32.4) indicates underweight patients were included in the weight losing intervention. The post- intervention BMI range (17.3-28.2) indicated that some could have gone to severer underweight. What your intervention ethical?

-your title and introduction do not indicate if were going to study Homeostatic model

assessment of Insulin resistance (HOMA-IR); HOMA-β: Homeostatic model assessment of

β-cell function

Discussion:

-revisit the stamen “To our knowledge, this is the first study to assess T2D remission in terms of achieving a normal glucose profile post-lifestyle intervention”

Conclusion

-Conclude on insulin resistance and B-cell function

6. PLOS authors have the option to publish the peer review history of their article (what does this mean?). If published, this will include your full peer review and any attached files.

Reviewer #1: No

Reviewer #2: No

---

## [Author Response · Author response to Decision Letter 0]

13 Jan 2024

RESPONSE TO REVIEWERS

We sincerely thank the editor and the reviewers for their valuable and insightful comments; they helped improve the quality of our manuscript. Please find below the detailed response to the reviewers’ comments for your kind perusal and necessary consideration. 

Additional Editor Comments:

This study addresses a crucial aspect of diabetes management — the ability to achieve normal glycaemic profiles post-remission through lifestyle interventions. This is a significant area of research with potential implications for diabetes care. Additionally, the intervention includes various elements like diet, exercise, psychological support, and medical management, which collectively contribute to a holistic treatment approach. Furthermore, the authors have collected a wide range of data points, including HbA1c levels, insulin resistance, β-cell function, and anthropometric measures, providing a robust basis for analysis. Despite all these pros, this manuscript could be improved by considering the following.

Comment 1. The study does not isolate the effects of individual components of the lifestyle intervention, which could be insightful for understanding which elements are most effective. Discussing this could also improve the manuscript.

Response: Thank you for your insightful comment. Due to the holistic nature of the intervention, we were unable to identify the isolated effect of individual components (Diet, Exercise, Psychological and Medical interventions) on T2D remission. We have added the same to the limitation section of the revised manuscript as below-

“Additionally, due to the holistic nature of the intervention with many factors at play, we could not study the isolated effect of any one component. Despite being a limitation, our research highlights the significant effectiveness of the lifestyle management program in achieving a better outcome in T2D management compared to a single intervention. The importance of holistic lifestyle intervention has been widely reported in several other studies, which highlights the importance of our findings in contributing to the body of knowledge in T2D management.” 

Comment 2. While the authors have reported statistical significance, the effect sizes of the interventions could be more explicitly stated to understand the magnitude of the changes observed.

Response: Thank you for your feedback on this. As per your suggestion, we have now incorporated pre-post values in Tables nos. 1 and 2 to highlight the effect size and the magnitude of the changes observed. 

Comment 3. The introduction could more explicitly state the study’s hypotheses or research questions.

Response: Thank you for your valuable suggestion. As suggested, we have revised the introduction to clearly state our hypothesis and research question as below-

“The current study aimed to investigate the association between OGTT clearance and T2D remission in patients who achieved remission after attending an online one-year lifestyle intervention program. The study hypothesis was that T2D-diagnosed patients in remission post-lifestyle intervention may be able to attain a normal glycemic profile through OGTT clearance, in addition to lowering HbA1c levels and improving insulin-related factors (HOMA-IR and HOMA-β).”

Comment 4. The manuscript should address any ethical considerations, particularly given the intervention's comprehensive nature and the potential impact on patients' lifestyles.

Response: As per the editor’s suggestions we have mentioned the implications of our interventions in the patients considering the ethical aspects, in the ‘Intervention section’ of the Methods as below-

“The ethical framework of our intervention emphasized the significance of individualized dietary modifications, customized exercise schedules, yoga, and stress management techniques, while simultaneously acknowledging the cultural nuances associated with dietary preferences and exercise choices. Our approach offered participants the flexibility to select from a range of options while still adhering to the primary objective of the intervention.” 

Comment 5. There may be an element of selection bias, as patients who opt into such a program might be more motivated or have different characteristics than the general diabetic population. This should be discussed.

Response: We acknowledge the element of selection bias, as individuals opting to participate in our lifestyle management program may exhibit higher motivation levels or possess different characteristics compared to the general diabetic population. We are thankful to the reviewer for prompting this discussion, which enhances the transparency and integrity of our research. Since this is one of the limitations of our study we have discussed it in the Discussion section as below-

“Another limitation is the selection bias due to the voluntary nature of participation in our lifestyle management program. Those who choose to participate may be more motivated or have unique traits when compared to the broader diabetes community. This self-selection could impact the external validity of our findings. Although the results of our lifestyle intervention program may not generalize to the broader population with diabetes, this does not diminish the importance of our findings within a specific group of participants.”

 

Reviewer Comments

Reviewer #1: I thank the authors for this relevant and informative study. However, the authors have overlooked some key issues as noted below.

Comment 1. The authors report that their study is a quasi-experimental pre-post-study design, however, some key data is not clearly depicted. The authors report baseline, post-intervention and post OGTT data for HOMA-IR and HOMA-B (table 1 and Figure 2), however given they report 3 specific outcome groups (cleared, IGT and not cleared) the baseline and post intervention data should also be grouped as such. There is quite a wide range in the measurements for both HOMA-IR and HOMA-B and it would be easier to discuss the observations noted if the data is grouped as such.

The authors also use post-intervention in table 2 is confusing. Do they mean post the 1 year intervention or post the 3 months of glycaemic-ladder’ training? The table heading should be more clear.

Response: Thank you for your comment. As per the suggestions we have revised and reported the baseline and post-intervention parameters in three groups based on OGTT status. We have also modified the tables and changed the headings. We have replaced Table 2 and Fig.2 with two new tables (Table 1 and 2) showing the comparison between and within the groups. To clarify, the three-month glycemic training is part of the one-year intervention for all those who are eligible. We have clarified this in the Methods section as below-

“As a part of the one-year lifestyle intervention program, upon achieving remission, participants were asked to voluntarily enrol for a three-month ‘glycemic-ladder’ training where they were gradually introduced to food items that were restricted during the intervention.”

Comment 2. The authors also report P values showing significance differences between the groups, however, these are only mentioned in the text and broad terms. The P values are not depicted in any of the figures, and the figures are also lacking in the number of participants for each observation. If the authors were also able to group the baseline and post-intervention data into the three observations noted they would be able to compare differences in these parameters that could predict the 3 observations in the OGTT.

Response: Thank you for the insightful comment on the data presentation. As per the suggestions, we have made the necessary changes and reported the exact p values and the participant numbers in the text as well as figures. We have also performed the analysis and compared them between the three groups based on OGTT results for all the parameters. 

Comment 3. Although the authors do report that the not-cleared group is small (n = 18), the results in this group warrant more exploration and discussion. In this group they observed improvements in HOMA-IR post-intervention however there was a decline in beta cell function as noted by HOMA-B measurements. In a previous diet intervention study of much lower duration (A Plant-Based Dietary Intervention Improves Beta-Cell Function and Insulin Resistance in Overweight Adults: A 16-week Randomized Clinical Trial, doi: 10.3390/nu10020189), improvement in beta-cell function was associated with improvements in HOMA-IR as is to be expected. The authors should discuss their unique results in more detail and offer plausible explanations for their observations.

Response: Thank you for your comment. We recognise the limited sample size of the "not cleared" group (n = 18) as well as the intriguing findings that need additional investigation. While there were improvements in HOMA-IR following intervention, there was a concurrent drop in beta-cell function as measured by HOMA-β. This is in contrast to the previously found connection between enhanced beta-cell activity and decreased insulin resistance in shorter-term research (doi: https://doi.org/10.3390/nu10020189). While our findings may appear to be exceptional, the literature does include examples of lifestyle treatments that do not uniformly improve both HOMA-IR and HOMA-B levels. Several studies have found individual responses to be variable, emphasising the variety in metabolic outcomes. As suggested, we have incorporated the possible explanation for this in the discussion section as below-

“Despite low numbers, a substantial reduction in insulin resistance (HOMA-IR) was observed post-intervention in the Not-cleared group, yet the results remained suboptimal (HOMA-IR=2.8); concurrently there was a decrease in HOMA-β, contradicting Kahleova et al.’s 2019 [31] reports of improved beta-cell function and fasting insulin sensitivity in a 16-week dietary intervention for overweight individuals without diabetes history. The intricate interplay of genetic and age-related factors, baseline metabolic status, and diabetes duration may contribute to the observed variations [13,32]. The small sample size poses a challenge in drawing definitive conclusions, due to the inherent variability in our study population for the Not-cleared group, emphasizing the need for cautious interpretation. Future research with a larger sample size could delve into the specific mechanisms underlying the observed changes, concerning HOMA-IR and HOMA-β.” 

 

Reviewer #2: General comment: You have a nice study if the following are included in your revision

Title: Okay

Abstract:

-Does not highlight type of lifestyle modification intervention that was offered

Response: As per the suggestion of the reviewer we have mentioned the type of lifestyle intervention offered to participants in the abstract as below-

“Four hundred participants who achieved remission during a one-year online structured lifestyle modification program, which included a plant-based diet, physical activity, psychological support, and medical management (between January 2021 and June 2022), and appeared for the OGTT were included in the study.”

Introduction:

-what was the problem here?

-What was you testing? Effectiveness of an intervention or suitability of OGTT in relation to HbA1c? state clearly

Response: Thank you for the valuable input. As suggested, we have revised the Introduction and added a clear statement of hypothesis along with clarity to what we aimed to study. 

Patient Enrolment:

-Why not call them participants?

Response: Thank you for your suggestion. We have replaced the word patient with participants in all relevant places.

-Do you think the nature of your sampling plan could have influenced your conclusion?

Response: Thank you for your insightful comment. We agree that our sampling approach has limitations and we have discussed this in detail in the limitation section of our manuscript as below-

“Another limitation is the selection bias due to the voluntary nature of participation in our lifestyle management program. Those who choose to participate may be more motivated or have unique traits when compared to the broader diabetes community. This self-selection could impact the external validity of our findings. Although the results of our lifestyle intervention program may not generalize to the broader population with diabetes, this does not diminish the importance of our findings within a specific group of participants.”

However, we want to highlight that our primary aim was to investigate the possibility of achieving normal glucose levels in T2D-diagnosed participants under remission. Despite the limitations, our study provides valuable insights into the relationship between OGTT clearance and remission and adds new data to the existing body of knowledge. In future studies planned, we will make sure to adopt randomization in our sampling. 

-do you think the potential participants who achieved remission differed from those who didn’t? and what about those who were not included in the final analysis? Would the results and hence conclusion been different if all who achieved remission had participated? Why?

Response: Thank you for your insightful comment. Based on our analysis for another paper on diabetes remission, we found significant differences between those who achieved remission vs those who did not especially in terms of weight loss and diabetes duration- those with weight loss of more than 15 kg and diabetes duration less than 6 years showed higher remission rates (unpublished data).

In the present study, since our focus was on individuals experiencing remission, the inclusion criteria mandated the enrollment of participants under remission. To answer your second query on those who enrolled but were not included in the final analysis, we conducted a preliminary analysis of Biochemical and Anthropometric parameters on the non-participant group (N=761) and notably, the results for non-participants also showed significant improvements for all the study parameters (Table given below) indicating a lesser chance of affecting our study outcome. 

Parameters Drop out Participant under remission (N=761) P values

 Pre-Intervention Post Intervention 

Weight (kg) 72 (65-81.3) 67.5 (60-75) <0.001

BMI (kg/m2) 25.8 (23.8-28.7) 23.8 (22.5-26.4) <0.001

HbA1c (mmol/mol) 51.9 (45.3- 61.7) 43.1 (38.7- 47.5) <0.001

Fasting Insulin (pmol/L) 49.8 (34.5-76.9) 41.4 (27.9 - 64.5) <0.001

Fasting Blood Sugar (mmol/L) 6.6 (5.7 – 7.7) 6.1(5.4 - 6.9) <0.001

Post Prandial Blood Sugar (mmol/L) 7.6 (6.1 – 9.8) 7.0 (5.8 -8.8) <0.001

HOMA-IR 2.6 (1.6 – 4.1) 1.9 (1.2 - 2.9) <0.001

HOMA-β 51.6 (31.1 – 88.7) 52.2 (32.1 -88.9) 0.314

We do acknowledge that a study including all participants in remission would have strengthened the overall validity and applicability of our findings. However, we had to consider the voluntary aspect of taking part in the current study.

Inclusion Criteria:

- did you account for potential contamination from alternative glucose-lowering therapies such as herbs which could have not been part of the intervention?

Response: Based on the eligibility criteria we considered only those participants who were off all glucose-lowering medicines including alternative medicine. We have added the same in the inclusion criteria for clarity as below-

“The primary inclusion criterion was remission defined as ‘HbA1c less than 48 mmol/mol without any glucose-lowering medicines (including alternative medicine) for at least 3 months…”

Intervention:

-What was the intervention? One year lifestyle modification versus 3-month glycaemic training.

Response: The intervention was an online one-year lifestyle modification programme that comprised dietary, exercise, psychological and medical management. The 3-month glycemic training was a part of the intervention for all those participants who underwent remission during the program. All those who achieved remission were invited to undergo the 3-month glycemic training during the one year they were with us. We have stated this more clearly in the Intervention Section of our manuscript as below-

“As a part of the one-year lifestyle intervention program, upon achieving remission, par

---

## [Decision Letter · Decision Letter 1]

4 Mar 2024

PONE-D-23-27928R1Oral Glucose Tolerance Test Clearance in Type 2 Diabetes Patients who underwent Remission following Intense Lifestyle modification: A Quasi-Experimental StudyPLOS ONE

Dear Dr. Kadam,

Thank you for submitting your manuscript to PLOS ONE. After careful consideration, we feel that it has merit but does not fully meet PLOS ONE’s publication criteria as it currently stands. Therefore, we invite you to submit a revised version of the manuscript that addresses the points raised during the review process.

address all the comments thoroughly This manuscript requires a minor revision==============================

We look forward to receiving your revised manuscript.

Kind regards,

Fredirick Lazaro mashili, MD, PhD

Academic Editor

PLOS ONE

Journal Requirements:

Additional Editor Comments:

Please address thoroughly the comments raised by one of the reviewers. Make sure you address them clearly as recommended

Reviewers' comments:

Reviewer's Responses to Questions

**Comments to the Author**

1. If the authors have adequately addressed your comments raised in a previous round of review and you feel that this manuscript is now acceptable for publication, you may indicate that here to bypass the “Comments to the Author” section, enter your conflict of interest statement in the “Confidential to Editor” section, and submit your "Accept" recommendation.

Reviewer #1: (No Response)

Reviewer #3: All comments have been addressed

2. Is the manuscript technically sound, and do the data support the conclusions?

Reviewer #1: Partly

Reviewer #3: Yes

3. Has the statistical analysis been performed appropriately and rigorously? 

Reviewer #1: No

Reviewer #3: Yes

4. Have the authors made all data underlying the findings in their manuscript fully available?

Reviewer #1: Yes

Reviewer #3: Yes

5. Is the manuscript presented in an intelligible fashion and written in standard English?

Reviewer #1: Yes

Reviewer #3: Yes

6. Review Comments to the Author

Reviewer #1: Comment 1: In the abstract as well as in the introduction and discussion the authors indicate that they they want to assess the association between OGTT clearance and diabetes remission. However, no statistical analyses were done to assess for association.

In the final paragraph the authors write "However, there is a lack of studies emphasizing the correlation between a normal glycemic profile in terms of OGTT clearance and T2D remission. The current study aimed to investigate the association between OGTT clearance and T2D remission in patients..." Neither association nor correlations were assessed in their results.

Comment 2: The methodology of the intervention is still unclear. The authors have responded that the glycemic ladder training occured in the course of the year of the lifestyle modification after which the OGTT was conducted. However how was this process influenced by the timetaken to attain remission? If a patient eg attained remission after 2 months VS 11 months, when were the glycemic ladder training and OGTT conducted for the two patients? what were the mean durations for remission for the three groups?

Comment 3: Results section is much improved. I would urge the authors to add a note at the bottom of table 2 explainig to the reader whether the more negative or positive the change for the parameters assessed reflects benefit.

Clinical characteristics such as mean age, gender etc of participants in the three groups is lacking and could shed light on the observations noted (doi: 10.2337/dc14-0874)

Reviewer #3: All the raised comments have been thoroughly and sufficiently addressed. The authors have made all necessary changes asked for by reviewers.

7. PLOS authors have the option to publish the peer review history of their article (what does this mean?). If published, this will include your full peer review and any attached files.

Reviewer #1: No

Reviewer #3: **Yes: **Fredirick mashili

---

## [Author Response · Author response to Decision Letter 1]

14 Mar 2024

RESPONSE TO REVIEWERS

We sincerely thank the editor and the reviewers for their valuable and insightful comments; they helped improve the quality of our manuscript. Please find below the detailed response to the reviewers’ comments for your kind perusal and necessary consideration. 

Additional Editor Comments:

Please address thoroughly the comments raised by one of the reviewers. Make sure you address them clearly as recommended.

Reviewer Comments

Reviewer #1: 

Comment 1. In the abstract as well as in the introduction and discussion the authors indicate that they they want to assess the association between OGTT clearance and diabetes remission. However, no statistical analyses were done to assess for association.

In the final paragraph the authors write "However, there is a lack of studies emphasizing the correlation between a normal glycemic profile in terms of OGTT clearance and T2D remission. The current study aimed to investigate the association between OGTT clearance and T2D remission in patients..." Neither association nor correlations were assessed in their results.

Response: Thank you for this comment. We accept our error in stating that we “aimed to investigate the association between OGTT clearance and T2D remission.” Since all participants in the study were already in remission, our study aimed to assess the possibility of OGTT clearance in patients who successfully achieved remission after one year of lifestyle intervention and to explore the factors associated with OGTT status. We have revised the same in our manuscript in the abstract as well as the introduction section, as below-

Abstract:

“The study investigates the factors associated with OGTT clearance in patients under type 2 diabetes remission.”

Introduction:

“However, there is a lack of studies exploring the possibility of achieving a normal glycemic profile in terms of OGTT clearance post-T2D remission. The current study aimed to investigate the factors associated with OGTT clearance in T2D patients who achieved remission after attending an online one-year lifestyle intervention program.”

Comment 2. The methodology of the intervention is still unclear. The authors have responded that the glycemic ladder training occured in the course of the year of the lifestyle modification after which the OGTT was conducted. However how was this process influenced by the timetaken to attain remission? If a patient eg attained remission after 2 months VS 11 months, when were the glycemic ladder training and OGTT conducted for the two patients? what were the mean durations for remission for the three groups?

Response: Thank you for your insightful comment. Remission was considered if the participant was able to maintain their HbA1c levels at less than 6.5% for at least 3 months after stopping all medications. Only 3 months after the complete cessation of medication, based on the HbA1c levels, the glycemic ladder training was initiated. Overall, the mean time taken to achieve remission before starting glycemic ladder training was 4.6 ± 1.9 months. The remission duration for the 3 groups was as follows: OGTT cleared group: 4.4 ± 2.0 months; IGT group: 4.7 ± 1.8 months; Not cleared group: 5.6 ± 2.5 months. Although the remission duration was the longest in the non-cleared group, we were unable to test for significance between groups due to the small sample size in the non-cleared group (n=18).

Most patients achieved remission within a one-year program. For those attaining remission towards the end of the program, glycemic ladder training support was extended through WhatsApp groups specially created for the study. The participants were allowed to submit their reports after the completion of the program. This was possible because of the personalized nature of the program. Therefore, if a participant attained remission early during the program, he started glycemic training earlier than someone who achieved remission later.

We have added the data to the results section of the manuscript as below-

“Overall, the mean time taken to achieve remission before starting glycemic ladder training was 4.6 ± 1.9 months.”

“The mean time taken to achieve remission before starting glycemic ladder training for the 3 groups was: OGTT cleared group, 4.4 ± 2.0 months; IGT group, 4.7 ± 1.8 months; Not cleared group, 5.6 ± 2.5 months.”

Comment 3: Results section is much improved. I would urge the authors to add a note at the bottom of table 2 explainig to the reader whether the more negative or positive the change for the parameters assessed reflects benefit.

Clinical characteristics such as mean age, gender etc of participants in the three groups is lacking and could shed light on the observations noted (doi: 10.2337/dc14-0874)

Response: Thank you for encouraging feedback on the Results section. As per the suggestions, we have revised and added a note to Table 2 to clarify whether the more negative or positive change in parameters reflects the benefit to enhance the reader's understanding as below- 

“The percentage decrease in weight, BMI, HbA1c, Fasting Insulin, Fasting Blood Sugar, PP-Blood Sugar, and HOMA-IR along with a percentage increase in HOMA-β indicates a positive effect and benefit to the participants.”

Additionally, to address the 2nd comment, we have added a new Table 1 to the Results section with sociodemographic characteristics (age, gender distribution, education, occupation, and marital status) for the three groups. The original figures 1A and 1B highlight other parameters such as diabetes duration and diabetes medication status (OHA and Insulin) that have been shifted after Table 1 to maintain continuity. The revised statement and table in the Results section have been incorporated as follows:

“The sociodemographic profiles of the participants in the three groups are presented in Table 1. All three groups had similar sociodemographic characteristics (P >0.1). Diabetes duration was significantly lower in the OGTT-cleared group than in the IGT group (p=0.046) (Fig 1A). Due to the low number of patients who did not clear the OGTT (N = 18), we could not statistically compare the outcome for this group. No significant difference in OGTT results was observed based on initial medication status (p>0.1) (Fig 1B).”

Table 1: Sociodemographic profile of participants in three outcome groups (N=400)

Parameters OGTT Status

 OGTT-cleared (N=207) IGT 

(N=175) Not-cleared 

(N=18)

Age (Years) 52.0±9.4 51.1±10.0 52.8±9.8

Gender Male 116 [56.0] 107 [61.1] 12 [66.6]

 Female 91 [44.0] 68 [38.9] 6 [33.4]

Education Graduate and below 109 [52.7] 98[56.0] 12 [66.7]

 Post-Graduate and above 98 [47.3] 77[44.0] 6 [33.3]

Occupation Employed 118 [57.0] 113 [64.6] 13 [70.6]

 Retired 28 [13.5] 19 [10.8] 3 [17.6]

 Homemaker 39 [18.8] 32 [18.3] 2 [11.8]

 Others a 22 [10.6] 11 [6.3] -

Marital Status

 Married 184 [88.8] 156 [89.1] 17 [94.1]

 Single b 23 [11.2] 19 [10.9] 1 [5.9]

Family History Yes 160 [77.1] 133 [76.0] 14 [76.5]

 No 47 [22.9] 42 [24.0] 4 [23.5]

Data are presented as mean ± standard deviation or frequency [percentage]; a Student/Preferred not to disclose; b Never married/separated/widowed; No significant differences were observed between the groups by Chi-square test (p>0.1)

Tables 1 and 2 are relabeled in Tables 2 and 3, respectively.  

Reviewer #2: All the raised comments have been thoroughly and sufficiently addressed. The authors have made all necessary changes asked for by reviewers.

Response: Thank you for your thorough review and constructive comments. We appreciate your valuable input that has undoubtedly improved the clarity and quality of our manuscript.

---

## [Decision Letter · Decision Letter 2]

12 Apr 2024

Oral Glucose Tolerance Test Clearance in Type 2 Diabetes Patients who underwent Remission following Intense Lifestyle modification: A Quasi-Experimental Study

PONE-D-23-27928R2

Dear Dr. Kadam,

We’re pleased to inform you that your manuscript has been judged scientifically suitable for publication and will be formally accepted for publication once it meets all outstanding technical requirements.

Kind regards,

Fredirick Lazaro mashili, MD, PhD

Academic Editor

PLOS ONE

Additional Editor Comments (optional):

All the comments have been sufficiently addressed

Reviewers' comments:

Reviewer's Responses to Questions

**Comments to the Author**

1. If the authors have adequately addressed your comments raised in a previous round of review and you feel that this manuscript is now acceptable for publication, you may indicate that here to bypass the “Comments to the Author” section, enter your conflict of interest statement in the “Confidential to Editor” section, and submit your "Accept" recommendation.

Reviewer #1: All comments have been addressed

Reviewer #3: All comments have been addressed

2. Is the manuscript technically sound, and do the data support the conclusions?

Reviewer #1: Yes

Reviewer #3: Yes

3. Has the statistical analysis been performed appropriately and rigorously? 

Reviewer #1: Yes

Reviewer #3: Yes

4. Have the authors made all data underlying the findings in their manuscript fully available?

Reviewer #1: Yes

Reviewer #3: Yes

5. Is the manuscript presented in an intelligible fashion and written in standard English?

Reviewer #1: Yes

Reviewer #3: (No Response)

6. Review Comments to the Author

Reviewer #1: I thank the authors for their patience during the review process and commend them on the much improved manuscript. The data presented is interesting and offers encouragement to both patients and clinicians that lifestyle modifications can be an avenue of management for DM and furthermore improve quality of care for patients.

Reviewer #3: All the previously raised comments have been sufficiently addressed. The authors have considered and adhered to all the journals requirements and standards.

7. PLOS authors have the option to publish the peer review history of their article (what does this mean?). If published, this will include your full peer review and any attached files.

Reviewer #1: No

Reviewer #3: **Yes: **Fredirick Mashili
